# Biomechanical Assessment of Red Blood Cells in Pulsatile Blood Flows

**DOI:** 10.3390/mi14020317

**Published:** 2023-01-26

**Authors:** Yang Jun Kang

**Affiliations:** Department of Mechanical Engineering, Chosun University, 309 Pilmun-daero, Dong-gu, Gwangju 61452, Republic of Korea; yjkang2011@chosun.ac.kr; Tel.: +82-62-230-7052; Fax: +82-62-230-7055

**Keywords:** rheological property, microfluidic pulsatile flow, blood viscoelasticity, RBC aggregation index, blood junction pressure, blood velocity, microscopic image intensity

## Abstract

As rheological properties are substantially influenced by red blood cells (RBCs) and plasma, the separation of their individual contributions in blood is essential. The estimation of multiple rheological factors is a critical issue for effective early detection of diseases. In this study, three rheological properties (i.e., viscoelasticity, RBC aggregation, and blood junction pressure) are measured by analyzing the blood velocity and image intensity in a microfluidic device. Using a single syringe pump, the blood flow rate sets to a pulsatile flow pattern (*Q_b_*[*t*] = 1 + 0.5 sin(2π*t*/240) mL/h). Based on the discrete fluidic circuit model, the analytical formula of the time constant (*λ_b_*) as viscoelasticity is derived and obtained at specific time intervals by analyzing the pulsatile blood velocity. To obtain RBC aggregation by reducing blood velocity substantially, an air compliance unit (ACU) is used to connect polyethylene tubing (i.d. = 250 µm, length = 150 mm) to the blood channel in parallel. The RBC aggregation index (AI) is obtained by analyzing the microscopic image intensity. The blood junction pressure (*β*) is obtained by integrating the blood velocity within the ACU. As a demonstration, the present method is then applied to detect either RBC-aggregated blood with different concentrations of dextran solution or hardened blood with thermally shocked RBCs. Thus, it can be concluded that the present method has the ability to consistently detect differences in diluent or RBCs in terms of three rheological properties.

## 1. Introduction

Red blood cells (RBCs) are highly deformable and constitute 40–50% of the blood volume. They carry oxygen into peripheral tissues and discharged carbon dioxide from them [1]. Rheological properties (including viscoelasticity, RBC aggregation, and deformability) are determined by RBCs and plasma. They significantly contribute to the varying blood flow in the capillary vessels [2]. Abnormal rheological changes induce blood clotting, irregular blood flow, and blockage of the blood vessels. The change of rheological properties of blood has been observed and correlated to diseases (such as lung cancer [3], sickle cell anemia [4,5,6], neurological disease [7], diabetes [8,9], chronic venous disease [10], atopic dermatitis [11], hypertension [12], and cardiovascular disease [13]). Based on the strong correlation between diseases and rheological properties [14], the mechanical properties of blood have been probed to monitor variations in the blood collected from patients. As rheological properties are substantially influenced by RBCs and plasma, it is necessary to decouple individual contributions. The measurement of multiple properties has been regarded as a critical issue for the effective detection of diseases.

Microfluidic devices that can provide in vitro biological microenvironments [15,16] or physiological capillary vessel structures [17] have been extensively adopted to measure rheological properties [18,19,20]. Several rheological properties, including viscosity [21,22,23,24,25], viscoelasticity [26], aggregation (or sedimentation) [27,28], deformability [29], and hematocrit [25,30,31], have been obtained by manipulating blood flows in microfluidic environments [32,33].

Among several rheological properties, blood viscosity, as a determinant factor of fluidic resistance, can be obtained by quantifying the blood flow in the capillary channel under a specific blood flow rate. As blood viscosity plays an important role in determining blood flow, several methodologies have focused on the quantification of blood flow, including interfacial relocation [34,35,36], liquid–air interface movement [25,37,38], droplet length [20,21], and membrane displacement [39,40,41]. Being influenced by several factors including RBC deformability, hematocrit, and plasma proteins, blood viscosity alone does not provide sufficient information for understanding the contribution of individual factors. RBC aggregation occurs at a sufficiently low shear rate or stasis rather than at higher shear rates. It is largely influenced by several factors such as the viscoelastic membrane, RBC morphology, and plasma proteins [42,43]. The RBC aggregation index (AI) is obtained by analyzing the image intensity of the blood flow in the capillary channel after stopping the blood delivery source [44,45,46]. Instead of the image intensity of RBC aggregation, a new term shear stress has been suggested as an alternative parameter for quantifying the degree of RBC disaggregation under continuous blood flow [47,48]. The shear stress estimated in a co-flowing channel has been used to quantify RBC sedimentation in the driving syringe [49]. According to previous studies, RBC aggregation has been quantified during stasis. It aims to detect changes in plasma proteins rather than the mechanical properties of RBC under continuous blood flow. As rheological properties (i.e., viscoelasticity and hematocrit) have a strong influence on RBC aggregation, they should be monitored by measuring RBC aggregation. Recently, based on parallel flow manipulation, RBC aggregation has been measured in the turn-off setting of the syringe pump. Subsequently, blood viscosity is obtained at the turn-on setting of the syringe pump [50]. Here, it is necessary to set a constant flow rate for the reference fluid, whose viscosity is specified in advance. Thus, two pumps are required to supply two fluids (i.e., the reference fluid and blood) with precise control of the flow rate. Furthermore, the previous method required blood flow to be stopped to measure RBC aggregation. To overcome the limitations of previous methods, it is necessary to measure RBC aggregation and rheological properties simultaneously, even without stopping blood flow. Moreover, because blood flow shows a pulsatile pattern in physiological environments [51,52,53], rheological properties should be monitored under continuous pulsatile blood flow. 

In this study, three rheological properties (viscoelasticity, RBC aggregation, and blood junction pressure) were simultaneously measured by analyzing the blood velocity and microscopic image intensity. To quantify viscoelasticity (i.e., time constant), the blood flow was set to pulsatile flow patterns with a single syringe pump. The time constant was then obtained by analyzing the blood velocity using a simple mathematical model. Next, to induce RBC aggregation at a low shear rate, an air compliance unit (ACU) is connected to the blood channel in parallel. As the blood velocity decreases substantially inside the ACU, RBC aggregation occurs. RBC aggregation was then obtained from the microscopic image intensity of the blood flow. Finally, the blood velocity within the ACU was used to estimate variations in blood junction pressure over time. 

Compared with previous studies, the present method can simultaneously measure three rheological properties (i.e., viscoelasticity, RBC aggregation, and blood junction pressure) under continuous pulsatile blood flow. By replacing the blood viscosity with the viscoelasticity, the syringe pump is reduced. A single syringe pump is sufficient to supply blood with a pulsatile profile. By connecting a simple ACU to the main channel in parallel, three rheological properties can be obtained at specific intervals.

## 2. Materials and Methods

### 2.1. Microfluidic Device and Experimental Setup

As shown in Figure 1(A-i), the microfluidic device suggested in a previous study [54] is modified to obtain three rheological properties under pulsatile blood flow. The device consists of one inlet and two outlets (a, b), a main channel with a uniform width (width = 1000 µm), and a side channel with a multistep width. The side channel consists of a narrow channel (width = 100 µm), wide channel (width = 1000 µm), and narrow channel (width = 100 µm) connected in series. The depth of all channels is fixed at 20 µm. Based on the protocol reported in a previous study [55], a polydimethylsiloxane (Sylgard 184, Dow Corning, Midland, MI, USA) microfluidic device is manufactured using a soft lithography technique.

The microfluidic device is then mounted on an inverted optical microscope (IX53; Olympus, Tokyo, Japan) equipped with a 4× objective lens (N.A. = 0.1). One polyethylene tubing (i.d. = 0.25 mm, length = 400 mm) is inserted into the inlet. The other tube (length = 200 mm) is fitted to the outlet (a). To repel air in the device and tubing, 1× phosphate-buffered saline (PBS) is injected through the tubing connected to the outlet (a). To embody the ACU, polyethylene tubing (i.d. = 250 µm, length = *L_ACU_*) is connected to the outlet (b). The open end of the tubing is completely closed using a steel bar, as shown in Figure 1(A-iv). The blood is loaded into a disposable syringe (~1 mL) and connected to the end of the polyethylene tubing connected to the inlet. The syringe is then installed in a syringe pump (neMESYS, Cetoni GmbH, Korbußen, Germany) aligned in the gravitational direction. As shown in Figure 1(A-ii), to induce the viscoelastic behavior of blood in the main channel, the flow rate is set to a pulsatile flow pattern of *Q_b_* (*t*) = *Q_α_* + *Q_β_* sin(2π*t*/*T*), where *Q_α_*, *Q_β_*, and *T* denote the mean flow rate, alternating flow rate, and period, respectively. As shown in Figure 1(A-iii), microscopic blood images are captured and saved using an image-acquisition system. The high-speed camera (FASTCAM MINI, Photron, Tokyo, Japan) is set to 2000 fps. With an external trigger, two sequential images are captured and recorded at intervals of 1 s during all the experiments. Figure 1(A-iv) shows a prototype of the proposed microfluidic system. Using a syringe pump, blood is supplied to the microfluidic channels. When the blood pressure increases significantly at the junction point, the blood moves toward the main channel and side channel simultaneously. As shown in the inset of Figure 1(A-iv), air compression inside the ACU contributes to sucking blood and 1× PBS as the vertical liquid column sequentially. An interface is then shown between air and 1× PBS inside the ACU.

### 2.2. Acquisition of Microscopic Image Intensity and Blood Velocity

As shown in Figure 1B, the image intensity of blood flow and blood velocity are obtained by digital image processing of microscopic images recorded during the experiments. First, a specific region of interest (ROI; 1 mm × 3 mm) is selected within the main channel. The image intensity of blood flow (<*I_m_*>) is obtained by averaging the image intensity of the blood flow over the ROI. Temporal variations in blood velocity are obtained with time-resolved micro-particle image velocimetry (micro-PIV) [56]. The interrogation window is set to 64 × 64 pixels. One pixel corresponded to 3.3 µm. The window overlap is set to 50%. Velocity fields are validated using a local median filter. According to a previous study [50,57], DOC (Depth-Of-Correlation) is calculated as DOC = 151.6 µm for the present imaging system. Because the DOC is much larger than the channel depth (*h* = 20 µm), it is inferred that the velocity fields are averaged and remain constant in the depth direction. Blood velocity (<*U_m_*>) is obtained by arithmetically averaging the velocity fields over the ROI [50]. Second, to quantify the image intensity and blood velocity in the side channel, a specific ROI (3 mm × 1 mm) is selected within the large-width section rather than the short-width section. Following the same procedures used for the quantification of image intensity and blood velocity within the main channel, image intensity (<*I_s_*>) and blood velocity (<*U_s_*>) are estimated by analyzing the microscopic image within the ROI. Finally, variations in four parameters (<*U_m_*>, <*U_s_*>, <*I_m_*>, and <*I_s_*>) are obtained under pulsatile blood flow. Here, blood (hematocrit [Hct] = 50%) is prepared by adding normal RBCs into a dextran solution (15 mg/mL). The blood flow rate is set to *Q_b_* (*t*) = 1 + 0.5 sin(2π*t*/240) mL/h. The length of the ACU is 150 mm (*L_ACU_* = 150 mm). Figure 1(C-i) shows temporal variations of <*U_m_*> and <*U_s_*>. Both velocities exhibit pulsatile flow patterns. <*U_s_*> is much smaller than <*U_m_*>. <*U_m_*> decreases slightly over time. <*U_s_*> also decreases gradually during the period when the pressure increased gradually inside the ACU. Figure 1(C-ii) shows the temporal variations in <*I_m_*> and <*I_s_*>. <*I_m_*> did not show substantial changes over time, even with pulsatile blood flow. Owing to RBC aggregation in the side channel, the <*I_s_*> shows periodic fluctuations (i.e., increase or decrease) over time. Two <Is> peaks occur when <*U_s_*> decreases, and its direction is reversed. The arrow mark (‘↑’) denotes the specific time of the two peaks of <*I_s_*>. In this study, <*U_m_*> and <*U_s_*> are used to quantify the viscoelasticity (i.e., time constant) and blood junction pressure, respectively. <*I_s_*> is used to obtain the RBC aggregation index (AI).

### 2.3. Analytical Formula of Time Constant of Blood Flow

As the blood flow was set to a pulsatile pattern, it was possible to obtain the time constant (i.e., viscoelasticity) of the blood flow in the main channel. As shown in Figure 2(A-i), blood is supplied in a pulsatile pattern (*Q_b_* [*t*] = *Q_α_* + *Q_β_* sin [2π*t*/*T*]) through the inlet. To embody the air compliance effect [58,59], an ACU is attached at the end of the side channel (i.e., outlet [b]). The blood junction pressure is denoted as *P_j_* at the junction (*j*) of both channels. Based on the pressure-driven flow relation (i.e., *ΔP* = *R* × *Q*, pressure drop = fluidic resistance × flow rate) [60], individual microfluidic channels filled with blood are modeled as fluidic resistance elements. As shown in Figure 2(A-ii), the main channel is modeled as two fluidic resistance elements connected in series (*R_a_*, *R_b_*). As shown in Figure 1(C-i), the blood velocity inside the side channel is much smaller than that in the main channel. Therefore, the fluidic resistance of the blood flow in the side channel is considered negligible. The ACU alone is modeled as a compliance element (*C_ACU_*). The blood flow rate controlled by the syringe pump is represented as an alternating source element (*Q_b_*). A mathematical model of the microfluidic channels is constructed using discrete fluidic circuit elements. When the mass conservation law is applied at the junction (*j*), the following equation is derived: (1)$$\frac{P_{j}}{R_{b}}+C_{ACU}\frac{d}{dt}\left(P_{j}\right)=Q_{b}.\;$$

The blood junction pressure (*P_j_*) is given by *P_j_* = *R_b_* × *Q_m_*, where *Q_m_* is the blood flow rate in the main channel. Substituting the formula for *P_j_* into Equation (1), the following differential equation is derived:(2)$$\lambda_{b}\frac{d}{dt}\left(Q_{m}\right)+Q_{m}=Q_{\alpha }+Q_{\alpha }\sin\left(\omega t\right).\;$$

*λ_b_* denotes the time constant of viscoelasticity and is expressed as *λ_b_* = *C_ACU_* × *R_b_*. *ω* was equal to *ω* = 2π/*T*. For a blood channel (i.e., width: *w*, depth: *h*, and length: *l*) with a low aspect ratio (AR) (AR = 20/1000)^61^, the analytical expression of the fluidic resistance (*R_b_*) was derived as approximately $R_{b}=\frac{12\;\mu_{b}l}{w\;h^{3}}$. Here, the *µ_b_* of blood. The steady solution of Equation (2) is derived as follows: (3)$$Q_{m}\left(t\right)=Q_{\alpha }+\frac{Q_{\beta }}{\sqrt{1+{\left(\omega \;\lambda_{b}\right)}^{2}}}\;\sin\left(\omega t-\theta_{0}\;\right).\;$$

The time delay (*θ_0_*) was expressed as $\theta_{0}=\tan^{-1}\left(\omega \lambda_{b}\right)$. As shown in Figure 1(C-i), the mean and alternating values of <*U_m_*> were obtained by conducting a curve-fitting technique of <*U_m_*> = *U_man_* + *U_alt_* sin (*ωt* − *θ_0_*).

**Figure 2 micromachines-14-00317-f002:**
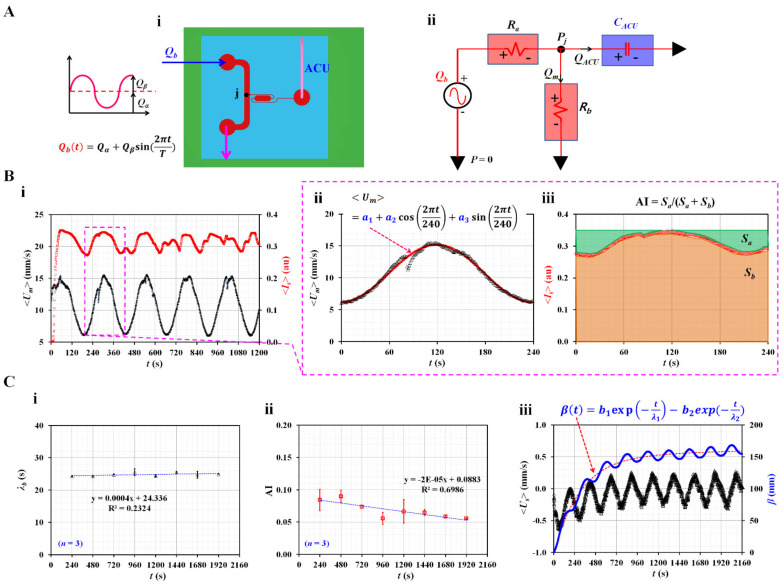
Mathematical representation of three rheological properties in terms of discrete fluidic circuit model. (**A**) Mathematical representation of microfluidic channels. (**i**) A top view of the microfluidic device. Blood flow rate set to pulsatile pattern. (**ii**) Mathematical representation of microfluidic circuit. A discrete fluidic circuit model consisted of flow rate source (*Q_b_*), fluidic resistance (*R_a_*, *R_b_*), and compliance (*C_ACU_*). (**B**) Quantification of time constant (*λ_b_*) and RBC aggregation index (AI). (**i**) Temporal variations of <*U_m_*> and <*I_s_*>. (**ii**) A curve-fitting procedure of <*U_m_*> for single period. The <*U_m_*> was then assumed as <*U_m_*> = *a_1_* + *a_2_* cos(2π*t*/240) + *a_3_* sin(2π*t*/240). (**iii**) Temporal variations of <*I_s_*> for single period. Based on temporal variations of <*I_s_*>, two parameters (*S_a_*, *S_b_*) were calculated as *S_a_* =$\;\int_{0}^{T}\left({<I_{s}>}_{max}-<I_{s}\left(t\right)>\right)dt,$ and *S_b_* = $\int_{0}^{T}<I_{s}\left(t\right)>dt$, respectively. RBC aggregation index (AI) was then obtained as AI = *S_a_*/(*S_a_* + *S_b_*). (**C**) Variation of three rheological properties at intervals of period. (**i**) Variation of *λ_b_* over period. (**ii**) Variations of AI over period. (**iii**) Temporal variations of <*U_s_*> and *β*. Based on the constitutive equation (i.e., *P_j_* = $\frac{1}{C_{ACU}}\int <U_{s}>dt),$ the *β* as new pressure-related variable was defined as *β* = −$\int <u_{s}>dt$. The *β* (*t*) was then assumed as *β* (*t*) = *b_1_* exp (−*t*/*λ_1_*) − *b_2_* exp (−*t*/*λ_2_*).

As the analytical expression of flow rate (*Q_m_*) was proportional to the blood velocity obtained with the micro-PIV technique (<*U_m_*>), the following relationship between *Q_m_* and <*U_m_*> is given as
(4)$$\frac{Q_{\alpha }}{U_{man}}=\frac{Q_{\beta }}{U_{alt}}\sqrt{1+{\left(\omega \lambda_{b}\right)}^{2}}.\;$$

Based on Equation (4), the *λ_b_* was derived as
(5)$$\lambda_{b}=\frac{T}{2\pi }\sqrt{{\left(\frac{Q_{\alpha }}{Q_{\beta }}\right)}^{2}{\left(\frac{U_{alt}}{U_{man}}\right)}^{2}-1}.\;$$

To obtain the time constant (*λ_b_*), it was necessary to obtain two velocity values: *U_man_* and *U_alt_*. Based on Equation (5), the time constant was then obtained at intervals of the period. 

### 2.4. Blood Preparation for Stimulating RBCs Sedimentation in the Driving Syringe

Concentrated RBC Bags (~320 mL) were purchased from the Gwangju–Chonnam Blood Bank (Gwangju, South Korea). First, to elevate RBC aggregation in RBCs, normal RBCs were added to several different types of dextran solutions. Five dextran solutions (C_dex_ = 15, 20, 40, 60, and 80 mg/mL) were diluted by dissolving dextran powder (Leuconostoc spp., MW = 450–650 kDa, Sigma-Aldrich, St. Louis, MO, USA) in 1× PBS. Unless otherwise stated, the Hct was set to 50%. Next, to harden the normal RBCs substantially, they were mixed with 1× PBS. Normal RBCs (approximately 5 mL) were loaded into a centrifuge tube. The tube was then exposed to 50 °C in a convection oven for 1 h [61]. As thermally shocked RBCs were collected by washing twice, test blood was prepared by adding hardened RBCs to a dextran solution (20 mg/mL).

## 3. Results and Discussion

### 3.1. Quantification of Three Rheological Properties in Pulsatile Blood Flow 

Three parameters (<*U_m_*>, <*U_s_*>, and <*I_s_*>) were used to obtain the three rheological properties. Among the four parameters (<*U_m_*>, <*U_s_*>, <*I_m_*>, and <*I_s_*>), as shown in Figure 1C, two (<*U_m_*>, <*I_s_*>) are redrawn from *t* = 0 s to *t* = 1200 s, as shown in Figure 2(B-i). To extract the mean value (*U_man_*) and alternating value (*U_alt_*) of <*U_m_*> during a single period (*T* = 240 s), as shown in Figure 2(B-ii), <*U_m_*> is replotted for 240 s from the specific time at which <*U_m_*> had the minimum value. To best fit the temporal variations in <*U_m_*> as periodic trigonometric functions, <*U_m_*> was assumed to be <*U_m_*> = *a_1_* + *a_2_* cos (2π*t*/240) + *a_3_* sin (2π*t*/240). Using the orthogonality relation of the periodic trigonometric function, three unknown constants (*a_1_*, *a_2_*, and *a_3_*) were calculated from the following relationship:(6)$$a_{1}=\frac{1}{240}\int_{0}^{240}<U_{m}>\;dt,\;$$
(7)$$a_{2}=\frac{2}{240}\int_{0}^{240}<U_{m}>\;\cos(\frac{2\pi t}{240})dt,\;$$
(8)$$a_{3}=\frac{2}{240}\int_{0}^{240}<U_{m}>\;\sin(\frac{2\pi t}{240})dt.\;$$

From Equations (6)–(8), three unknown constants were calculated: *a_1_* = 10.68, *a_2_* = −4.51, and *a_3_* = 0.11. *U_man_* and *U_alt_* were then calculated as $U_{man}=a_{1}=10.68$ and $U_{alt}=\sqrt{a_{2}{}^{2}+a_{3}{}^{2}}=4.51$. By inserting five parameters (i.e., *T* = 240 s, *Q_α_* =1 mL/h, *Q_β_* = 0.5 mL/h, *U_man_* = 10.68 mm/s, and *U_alt_* = 4.51 mm/s) into Equation (5), time constant was obtained as *λ_b_* = 24.18 s. To obtain RBC aggregation index (AI), as shown in Figure 2(B-iii), the <*I_s_*> was replotted for single period. <*I_s_*> exhibits pulsatile trends over time because <*U_m_*> varies in a pulsatile pattern. According to a previous study, blood flow should be suddenly stopped to measure RBC aggregation. Blood image intensity decreases or increases over time, depending on the optical system [45,54]. Owing to the difference in the blood flow profile (i.e., sudden stop flow or pulsatile flow), microscopic image intensity (<*I_s_*>) exhibited a substantially different trend over time. However, the present method adopted a similar protocol for calculating AI, as reported in a previous study [45,50]. Based on the temporal variations in <*I_s_*>, two parameters (*S_a_*, *S_b_*) were calculated as *S_a_* =$\;\int_{0}^{T}\left({<I_{s}>}_{max}-<I_{s}\left(t\right)>\right)dt,$ and *S_b_* = $\int_{0}^{T}<I_{s}\left(t\right)>dt$, respectively. Here, <*I_s_*>_max_ denotes the maximum value of <*I_s_*> in a single period. *S_a_* exhibited a unique contribution resulting from the RBC aggregation. In addition, *S_b_* was calculated to normalize AI as a dimensionless index. The RBC aggregation index (AI) is defined as AI = *S_a_*/(*S_a_* + *S_b_*). The two parameters were calculated *S_a_* = 7.879 and *S_b_* = 74.564. The RBC aggregation index was calculated as AI = 0.096. As shown in Figure 2(C-i,ii), *λ_b_* and AI are obtained at specific intervals. Figure 2(C-i) shows variations in *λ_b_* at certain intervals. Based on repetitive tests (*n* = 3), each AI was represented as the mean ± standard deviation. According to linear regression, the regression coefficient had a low value of R^2^ = 0.2324. From these results, it is evident that *λ_b_* remained unchanged over time. Figure 2(C-ii) shows the variations in AI over time. The AI tends to decrease significantly over time (R^2^ = 0.6986). This was due to RBC sedimentation in the driving syringe. While blood was supplied from the driving syringe into the microfluidic channel, the hematocrit of blood flow tended to increase over time [49]. The gradually increasing hematocrit contributed to hindering RBC aggregation. Blood junction pressure could be estimated from the constitutive relationship in the compliance element (i.e., $P_{j}=\frac{1}{C_{ACU}}\int_{0}^{t_{s}}<U_{s}>dt$). However, it was impossible to extract the compliance (*C_ACU_*) from the formula of the time constant (i.e., *λ_b_* = *C_ACU_* × *R_b_*) because fluidic resistance (*R_b_*) was not given during each experiment. As shown in Figure 1B, blood flows from right to left within the ROI of the side channel. To represent the junction pressure without information on *C_ACU_*, a new variable (*β*) was defined as *β* = $-\int <u_{s}>dt$. Figure 2(C-iii) shows the temporal variations in <*U_s_*> and *β*. *β* tends to increase gradually over time and exhibits a periodic pulsatile pattern. To analyze *β* quantitatively, *β* (*t*) was best fitted as *β* (*t*) = *b_1_* exp (–*t*/*λ_1_*)–*b_2_* exp (–*t*/*λ_2_*). According to the nonlinear regression analysis, four unknown constants (i.e., *b_1_*, *b_2_*, *λ_1_*, and *λ_2_*) were obtained as *b_1_* = 151.8, *b_2_* = 153.9, *λ_1_* = 45,413.26, and λ_2_ = 306.18. Thus, *b_1_* and *b_2_* have similar values (i.e., *b_1_* ≈ *b_2_*). *λ_2_* could be considered a vital parameter because *λ_1_* is much longer than *λ_2_* (i.e., *λ_1_*
≫
*λ_2_*).

### 3.2. Contributions of Three Vital Factors (i.e., Flow Profile, Air Cavity, and Hematocrit) to Rheological Properties

Three promising factors (blood flow profile, air cavity size in the ACU, and hematocrit) that might influence the performance of the present method were selected. Their contributions were evaluated quantitatively by measuring the image intensity and velocity of blood in both channels. Here, the same blood, which was prepared by suspending normal RBCs in dextran solution (15 mg/mL), was used for the following experiments. 

First, three types of blood rate patterns (constant, pulsatile, and square-wave) were selected to evaluate their contributions to variations in velocity, image intensity, and blood junction pressure. The length of the ACU was set to 150 mm (*L_ACU_* = 150 mm) and hematocrit was set to 50%. Figure 3A shows temporal variations in blood velocities (<*U_m_*> and <*U_s_*>), blood junction pressure (*β*), image intensities (<*I_m_*> and <*I_s_*>) under a constant flow rate (*Q_b_* = 1 mL/h). The <*U_m_*> remains unchanged over time. However, <*U_s_*> decreases gradually over time and is approximately zero at 1200 s. To represent the BP, *β* is obtained by integrating <*U_s_*> over time. It exhibits a gradual increase for up to 1200 s. Thereafter, it does not exhibit a substantial change over time. Based on the curve-fitting technique, three unknown constants of the regression formula of *β* are obtained: *b_1_* = 120.0 ± 9.7, *b_2_* = 115.8 ± 6.9, and *λ_2_* = 258.6 ± 6.1 (*n* = 3). The image intensity in the main channel (<*I_m_*>) remains unchanged over time. However, <*I_s_*> decreases substantially from 132 to 1148 s because <*U_s_*> tends to decrease over time. It is estimated that RBC aggregation contributes to a decrease in the image intensity (<*I_s_*>) in the side channel. Thereafter (*t* > 1148 s), as <*U_s_*> decreases to nearly zero, the variations in <*I_s_*> are influenced by random flows in the side channel (i.e., increase–decrease–increase–constant). Based on the temporal variations in <*I_s_*> from *t* = 132 s to *t* = 1148 s, AI was then obtained as AI = 0.14 ± 0.01 (*n* = 3). 

Figure 3B shows temporal variations in blood velocities (<*U_m_*> and <*U_s_*>), blood junction pressure (*β*), image intensities (<*I_m_*> and <*I_s_*>) under pulsatile flow rate (i.e., *Q_b_* = 1 + 0.5 sin [2π*t*/240] mL/h). Both the blood velocities exhibited pulsatile patterns over time. Based on the curve-fitting procedure, three unknown constants of the regression formula of *β* were obtained as *b_1_* = 155.2 ± 4.1, *b_2_* = 161.3 ± 4.2, and *λ_2_* = 280.9 ± 4.6 (*n* = 2). AI tended to decrease gradually from 0.099 to 0.04, with an increase in period. Figure 3C showed temporal variations of blood velocities (<*U_m_*>, and <*U_s_*>), blood junction pressure (*β*), and image intensities (<*I_m_*>, and <*I_s_*>) under square-wave flow rate (i.e., *Q_max_* = 1.5 mL/h, *Q_min_* = 0.5 mL/h, and *T* = 240 s). Both blood velocities exhibited square wave patterns over time. By conducting regression analysis, three unknown constants of regression formula of *β* were obtained as *b_1_* = 134.2 ± 1.1, *b_2_* = 134.3 ± 0.3, and *λ_2_* = 256.3 ± 12.8 (*n* = 2). AI tended to decrease gradually from 0.105 to 0.028 with an increase in the period. From the results, the AI obtained at a constant flow rate had a maximum value when compared with the remaining profiles. However, at alternating blood flow rates, variations in AI could be obtained at specific time intervals. Consequently, it tended to decrease with increasing period. In addition, based on the curve-fitting procedure, the constant and square-wave profiles had similar value of *λ_2_*. The *λ_2_* of the pulsatile profile was longer than that of the square-wave profile. Thus, the AI and *β* were substantially influenced by the blood flow rate profile. Among the three profiles, for convenience, to obtain rheological properties at specific time intervals, the blood flow rate was set to the pulsatile profile (i.e., *Q_b_* = 1 + 0.5 sin [2π*t*/240] mL/h) in following experiments.

Second, as it was expected that the air compliance unit (ACU) would influence blood flow in the side channel, its contribution was evaluated by varying the length of the ACU (*L_ACU_*). 

According to previous studies [59,62,63,64], the compliance element has been used to reduce unstable flow in microfluidic channels. However, in this study, ACU was used to substantially decrease blood flow and quantify blood junction pressure simultaneously. Figure 4A shows the temporal variations in <*U_m_*> and *β* with respect to *L_ACU_* = 50, 100, and 200 mm. The *β* tends to increase with increasing ACU length. According to a previous study, a longer air cavity contributed to increasing air compliance (*C_ACU_*) [62]. To maintain the same blood junction pressure, this result was regarded as reasonable because *β* should be increased at longer ACU lengths. Figure 4B shows the temporal variations in <*I_s_*> with respect to *L_ACU_* = 50, 100, and 200 mm. From the results, it was confirmed that <*I_s_*> exhibited consistent periodic patterns at larger ACU lengths (i.e., *L_ACU_* = 100 or 200 mm). Figure 4C shows the quantification of three parameters (*β*, *λ_b_*, and AI) with respect to *L_ACU_*. Based on the curve-fitting technique, three unknown constants (*b_1_*, *b_2_*, and *λ_2_*) of the regression formula for *β* were obtained with respect to *L_ACU_*. As shown in Figure 4(C-i), *b_1_* and *b_2_* exhibit similar trends and increase gradually with respect to *L_ACU_*. Furthermore, *λ_2_* tends to increase substantially with respect to *L_ACU_*. Figure 4(C-ii) shows the temporal variations in the time constant (*λ_b_*) with respect to *L_ACU_*. *λ_b_* tends to increase with increasing ACU length. Because *λ_b_* is proportional to *C_ACU_* (i.e., *λ_b_* = *R_b_* × *C_ACU_*), the experimental results exhibit consistent trends with respect to *L_ACU_*. However, *λ_b_* does not show substantial changes at the intervals of the period. Figure 4(C-iii) shows the temporal variations in AI with respect to *L_ACU_*. AI exhibits large fluctuations at lower ACU lengths (i.e., *L_ACU_* ≤ 100 mm). The fluctuation of AI tends to decrease at larger ACU lengths (*L_ACU_* ≥ 150 mm). It tends to decrease substantially at larger ACU lengths. The AI decreases gradually at intervals of the period. Based on these results, the length of the ACU was set to 150 mm in the following experiments.

Third, the contributions of the hematocrit to the three rheological properties were evaluated in pulsatile blood flow. As hematocrit plays an important role in blood flow, it was necessary to evaluate the rheological properties with respect to Hct = 30, 40, and 50%.

Figure 5(A-i) shows the temporal variations of <*U_m_*> with respect to hematocrit. Lower hematocrit (Hct = 30%) has higher value of blood velocity within less than two periods. Blood velocity of blood (Hct = 30%) increases largely from the seventh period. Blood velocity (Hct = 40%) increases substantially from the eighth period. As RBC sedimentation occurred in the driving syringe, diluent and RBCs were separated into the syringe. Both were supplied sequentially into the main channel. After a certain period, pure liquid (i.e., diluent) came into the main channel after all RBCs were already supplied into the microfluidic channel [65]. Owing to a feature of micro-PIV technique, the velocity of diluent is higher than that of blood with RBCs [66]. Figure 5(A-ii) shows temporal variations of *β* with respect to Hct. The *β* of blood (Hct = 50%) increases gradually up to the third period. After that period, it shows pulsatile patterns over time without an additional increase in the average value. The remaining two hematocrits of blood (Hct = 30 and 40%) do not exhibit substantial difference of *β* for up to the fifth period. After that period, the *β* exhibits substantial difference because of RBC sedimentation in the driving syringe. Figure 5(A-iii) shows temporal variations of <*I_m_*> with respect to Hct. The <*I_m_*> of 50% hematocrit is higher than *that* of Hct = 30 or 40%. RBC sedimentation in the driving syringe contributes to increasing <*I_m_*> gradually over time with respect to Hct = 30 or 40%. When diluent comes into the main channel, <*I_m_*> decreases substantially. Thus, the corresponding *<I_m_>* of Hct = 30% and Hct = 40% decreases abruptly at *t* = 1573 (Hct = 30%) and *t* = 1927 s (Hct = 40%), respectively. The difference between Hct = 30% and Hct = 40% can be detected effectively with sudden change of <*I_m_*> rather than steady value of <*I_m_*>. However, the <*I_m_*> of Hct = 50% remains constant over time. Figure 5(A-iv) shows temporal variations of <*I_s_*> with respect to Hct. Within the second period, the <*I_s_*> shows substantial difference with respect to Hct. Thereafter, the <*I_s_*> does not show noticeable difference with respect to Hct. Based on the temporal variation of <*I_m_*> over the period, temporal variations of *λ_b_* are obtained with respect to Hct. Figure 5(B-i) shows temporal variations of *λ_b_* with respect to Hct. The *λ_b_* of Hct = 30% is higher than that of Hct = 40 or 50%. According to the formula *λ_b_* = *R_b_* × *C_ACU_*, it was expected that the *C_ACU_* of 30% hematocrit was larger than that of 40 or 50% because viscosity of 30% hematocrit is higher than that of 40 or 50%. Thus, as compliance had reciprocal relation to elasticity, elasticity of 30% hematocrit is lower than that of 40 or 50%. For this reason, the result shows consistent trends when compared with previous study^62^. The *λ_b_* of Hct = 50% remains consistent over time. Figure 5(B-ii) shows temporal variations of AI with respect to Hct. The AI of Hct = 30 or 40% exhibits large fluctuations when compared with that of Hct = 50%. For consistent measurement of the present method, hematocrit of blood was set to 50% in the following experiments. 

### 3.3. Quantification of Three Rheological Properties for Aggregated Blood and Hardened Blood

As a demonstration, the present method was applied to detect differences in diluent and RBC flexibility. To increase RBC aggregation in normal RBCs, blood samples (Hct = 50%) were prepared by adding normal RBCs to a specific concentration of dextran solution (C_dex_) (C_dex_ = 20, 40, 60, and 80 mg/mL). Additionally, to harden normal RBCs under thermal shock conditions, normal RBCs were exposed to a higher temperature of 50 °C for 1 h. Hardened blood (Hct = 50%) as test blood was then prepared by adding thermally shocked RBCs into a specific concentration of dextran solution (20 mg/mL).

First, the contribution of the diluent (i.e., dextran solution) to the rheological properties was evaluated with respect to C_dex_. Three types of blood tests (*n* = 3) were conducted to confirm the repeatability of the present method. All tests were completed in three days. As shown in Figure A1 (Appendix A), by integrating <*U_s_*> over time, the temporal variations in *β* are obtained with respect to C_dex_. The results indicate that *β* increases at higher concentrations of dextran solution. In addition, four snapshots showing the volume of blood filled in the ACU were captured at the end of the blood delivery (*t* = 2160 s). The blood volume in the ACU increases at higher concentrations of dextran solution. Based on the temporal variations of *β*, as shown in Figure A1 (Appendix A), nonlinear regression is conducted to extract four unknown constants. Because *b_1_* and *b_2_* have very similar values (i.e., *b_1_* ≈ *b_2_*), *b_1_* is adopted to represent trends of *β* quantitatively. Figure 6A shows the variations in *b_1_* and *λ_2_* with respect to *C_dex_*. According to the results, both constants (*b_1_* and *λ_2_*) tend to increase with respect to C_dex_. The trends of both the constants are consistent for the three types of blood tests. Thus, it is inferred that the dextran solution played an important role in increasing blood junction pressure substantially. Based on the temporal variations in <*U_m_*>, the time constant of blood flow (*λ_b_*) was obtained at different time intervals. Figure 6B shows temporal variations in *λ_b_* with respect to C_dex_. The *λ_b_* increases at higher concentrations of dextran solution. In addition, blood samples with a higher concentration of dextran solution exhibit a substantial increase in *λ_b_* over time. Here, it is expected that RBC sedimentation in the driving syringe will contribute to increasing the hematocrit of the blood flow in the blood channel. As the hematocrit contribute to increasing blood viscosity, it is reasonable that *λ_b_* tended to increase over time. As shown in Figure A2 (Appendix A), the AI is obtained at intervals of the period by analyzing the temporal variations of <*I_s_*> with respect to C_dex_. Interestingly, two peaks of <*I_s_*> are detected when <*U_s_*> tends to decrease gradually, and its flow direction is reversed. Two peaks of <*I_s_*> are denoted as an arrow mark (‘↑’) with respect to C_dex_. Based on the specific time when two peaks occur, the AI is calculated from one period ago. Figure 6C shows the temporal variations in AI with respect to C_dex_. The AI tends to increase up to C_dex_ = 40 mg/mL. Above C_dex_ = 40 mg/mL, AI exhibits fluctuations with respect to C_dex_ = 60 and 80 mg/mL. Because of the ESR in the driving syringe, the AI varies with the elapsed period. The experimental results show that the three rheological properties exhibit substantial differences with respect to C_dex_.

Second, the present method was used to detect differences between normal RBCs and hardened RBCs in terms of the three rheological properties. Control blood (Hct = 50%) and test blood (Hct = 50%) were prepared by adding normal RBCs and thermally shocked RBCs to the same concentration of dextran solution (20 mg/mL). Figure 7(A-i) shows the temporal variations in *β* with respect to the control blood and test blood. The *β* of the test blood is lower than that of the control blood. The results were contrary to expectations and interesting. According to a previous study [67], hardened RBC largely contributed to decreasing compliance when compared with control blood. However, as the C*_ACU_* was not specified, it was impossible to obtain the blood junction pressure. Variations in *β* still showed a limitation in the representation of blood junction pressure when the elasticity of RBC was varied. To determine the reason quantitatively, it is necessary to conduct additional studies in the future. To analyze *β* of both blood samples quantitatively, based on the curve-fitting technique of *β*, unknown constants of a curve-fitting formula (i.e., *b_1_*, *λ_2_*) were obtained. As shown in Figure 7(A-ii), variations in *b_1_* and *λ_2_* are plotted with respect to the test and control blood. The *λ_2_* does not exhibit a substantial difference. The *b_1_* of test blood is lower than that of the control blood. Based on the temporal variations of <*U_m_*> and <*I_s_*>, *λ_b_* and AI are obtained at intervals of a period. Figure 7B shows the temporal variations in *λ_b_* and AI with respect to the control blood and test blood. The *λ_b_* value of the test blood is higher than that of the control blood. Compared with a previous study [61], the results obtained by the present method were comparable. Additionally, the AI of the test blood was lower than that of control blood. Therefore, heat-shocked RBCs contributed to decreasing AI and increasing *λ_b_* when compared with normal RBCs. It is to be noted that the *β* value of the test blood was lower than that of control blood. 

According to the previous studies, the author group suggested quantification methods of blood biomechanical properties in a microfluidic channel. For the quantification of blood viscosity, two highly expensive and bulky-sized syringe pumps were used to obtain supply blood and reference fluid, respectively [34,35]. Next, for quantification of RBC aggregation, blood syringe pump was periodically turned off to stop blood flow [44]. Most of all, as RBC aggregation depended on hematocrit or diluent, it was necessary to monitor variations of hematocrit or diluent in terms of blood viscosity or blood pressure. In other words, most of the previous studies were demonstrated with two syringe pumps. The present study was aimed at reducing the number of syringe pump from two to one. Blood viscosity was replaced by blood viscoelasticity (i.e., a time constant). With a single syringe pump, blood flow was set to pulsatile pattern. Three biomechanical properties (i.e., RBC aggregation, time constant, and blood junction pressure) were then simultaneously quantified under pulsatile blood flows. As a key idea, blood flow was stopped in the side channel connected to air compliance unit. Time constant was obtained by analyzing blood velocity in the main channel. Additionally, blood junction pressure was obtained by integrating blood velocity in the side channel. In principle, all formulas used or derived in the study had similar expression when compared with the previous studies. The present results showed very similar trends to the previous results. However, blood hematocrit had a strong influence on blood velocity obtained with a micro-PIV technique [66]. Furthermore, blood junction pressure was obtained as *β* because compliance constant was not available [67]. Accurate measurement of blood velocity as well as compliance constant will be required for improving the performance of the present method. 

From the experimental results, it can be concluded that the present method has the ability to detect differences in diluent or RBCs in terms of the three discussed rheological properties. A limitation of the present method is that it still requires laboratory-equipped facilities, including a microscope, high-speed camera, and syringe pump. 

## 4. Conclusions

In this study, three rheological properties (viscoelasticity, RBC aggregation, and blood junction pressure) were measured by analyzing the blood velocity and image intensity. The blood flow was set to a pulsatile flow pattern with a single syringe pump. First, based on the discrete fluidic circuit model, the analytical formula of the time constant (*λ_b_*) as a function of viscoelasticity was derived and estimated by analyzing the blood velocity. Next, to reduce the blood velocity substantially, an air compliance unit (ACU) was connected to the blood channel in parallel. The RBC aggregation index (AI) was obtained by analyzing the microscopic image intensity. Finally, based on the constitutive relation (i.e., $P_{j}=\frac{1}{C_{ACU}}\int_{0}^{t_{s}}<U_{s}>dt$), the blood junction pressure (*β*) was obtained by integrating the blood velocity within the ACU. From quantitative studies of three vital factors (i.e., blood flow rate profile, length of ACU, and hematocrit), the blood flow rate was set to a pulsatile profile (i.e., *Q_b_*[*t*] = 1 + 0.5 sin(2π*t*/240) mL/h). The length of the ACU was set as 150 mm (i.e., *L_ACU_* = 150 mm). The hematocrit of the blood was set to 50%. As a demonstration, the proposed method was applied to detect RBC aggregation-enhanced blood and heat-shocked RBCs. Consequently, the present method has the ability to detect differences in diluent or RBCs in terms of the three rheological properties. In the near future, to adopt the present method in clinical settings, it will be necessary to replace the bulky and expensive instruments with a facile and inexpensive system (i.e., imaging acquisition and blood delivery).

## Figures and Tables

**Figure 1 micromachines-14-00317-f001:**
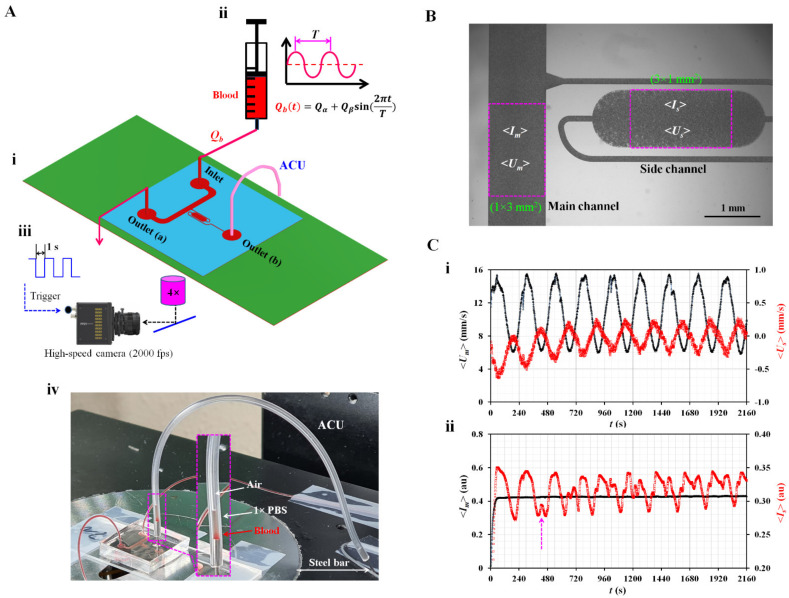
Proposed method for quantifying rheological properties in pulsatile blood flows. (**A**) Schematic diagram of a proposed method. (**i**) Microfluidic device with one inlet, two outlets (a, b), a main channel, and a side channel. Air compliance unit (ACU) was connected to outlet (b). The flow rate set to *Q_b_* (*t*) = *Q_α_* + *Q_β_* sin(2π*t*/*T*). Here, the *T* denoted period. (**iii**) Microscopic image acquisition system. (**iv**) A protype of a suggested microfluidic system. (**B**) Quantification of four parameters of blood flow in main and side channels (i.e., <*U_m_*>, <*I_m_*>, <*U_s_*>, and <*I_s_*>). (**C**) Quantification of four parameters for dextran-included blood (i.e., normal RBCs suspended into dextran solution [15 mg/mL], and Hct = 50%). (**i**) Temporal variations of <*U_m_*> and <*U_s_*>. (**ii**) Temporal variations of <*I_m_*> and <*I_s_*>.

**Figure 3 micromachines-14-00317-f003:**
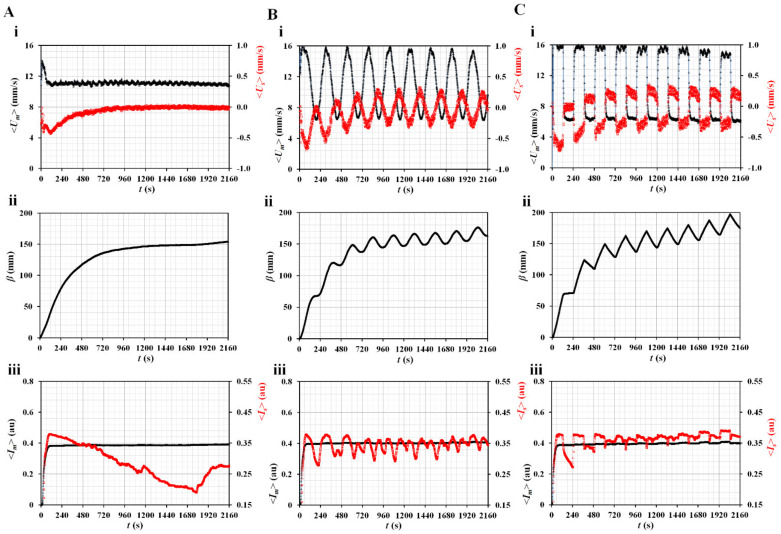
Contributions of a flow rate profile to variation of velocity, image intensity, and pressure in the side channel. (**A**) Temporal variations of (**i**) blood velocities (<*U_m_*>, <*U_s_*>), (**ii**) *β*, (**iii**) image intensities (<*I_m_*>, <*I_s_*>) under constant flow rate (*Q_b_* = 1 mL/h). (**B**) Temporal variations of (**i**) blood velocities (<*U_m_*>, <*U_s_*>), (**ii**) *β*, (**iii**) image intensities (<*I_m_*>, <*I_s_*>) under pulsatile flow rate (*Q_b_* = 1 + 0.5 sin (2π*t*/240) mL/h). (**C**) Temporal variations of (**i**) blood velocities (<*U_m_*>, <*U_s_*>), (**iii**) *β*, and (**iii**) image intensities (<*I_m_*>, <*I_s_*>), and under square-wave flow rate (*Q_max_* = 1.5 mL/h, *Q_min_* = 0.5 mL/h, and *T* = 240 s).

**Figure 4 micromachines-14-00317-f004:**
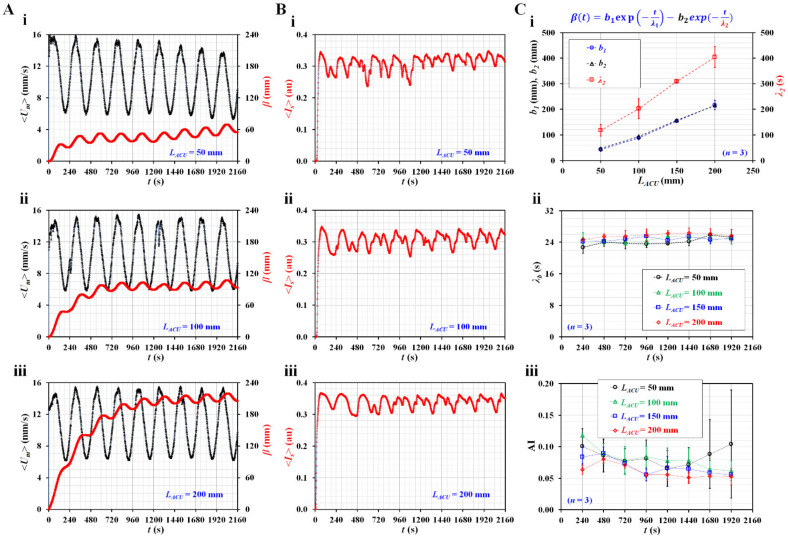
Contributions of air compliance unit (ACU) to three rheological properties in pulsatile blood flow rate. (**A**) Temporal variations of <*U_m_*> and *β* with respect to *L_ACU_* = 50, 100, and 200. (**B**) Temporal variations of <*I_s_*> with respect to *L_ACU_*. (**C**) Quantifications of three parameters (i.e., *β*, λ_b_, and AI) with respect to *L_ACU_*. (**i**) Variations of *b_1_*, *b_2_*, and *λ_2_* with respect to *L_ACU_*. (**ii**) Temporal variations of time constant (*λ_b_*) with respect to *L_ACU_*. (**iii**) Temporal variations of AI with respect to *L_ACU_*.

**Figure 5 micromachines-14-00317-f005:**
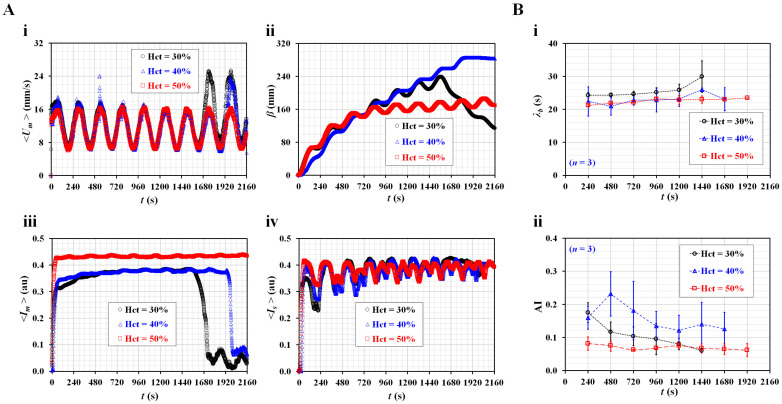
Contribution of hematocrit to rheological properties in pulsatile blood flow. (**A**) Quantification of four parameters (i.e., <*U_m_*>, *β*, <*I_m_*>, and <*I_s_*>) with respect to Hct = 30, 40, and 50%. Temporal variations of (**i**) <*U_m_*>, (**ii**) *β*, (**iii**) <*I_m_*>, and (**iv**) <*I_s_*> were obtained with respect to Hct. (**B**) Quantification of two properties (*λ_b_*, AI) with respect to Hct. Temporal variations of (**i**) *λ_b_*, and (**ii**) AI were obtained with respect to Hct.

**Figure 6 micromachines-14-00317-f006:**
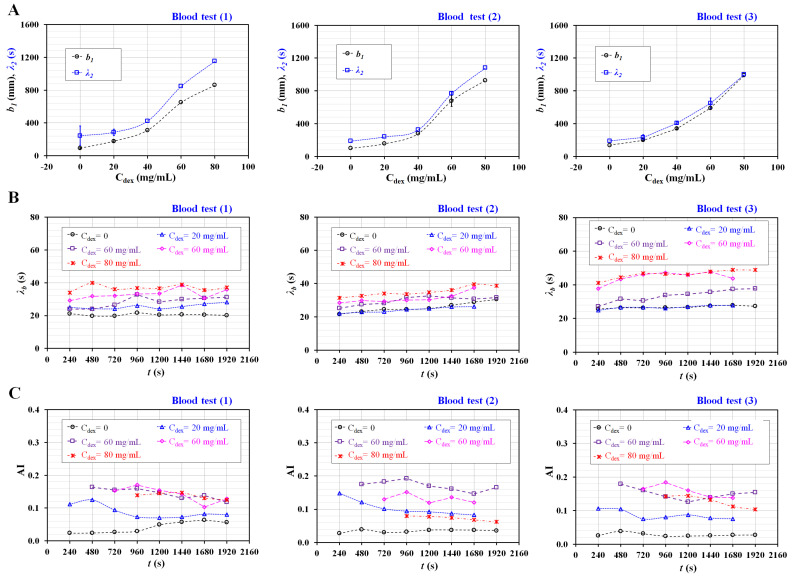
Quantification of three rheological properties for RBC aggregation-enhanced blood. Blood (Hct = 50%) was prepared by adding normal RBCs into a specific concentration of dextran solution (C_dex_) (C_dex_ = 20, 40, 60, and 80 mg/mL). Three kinds of blood test (*n* = 3) were conducted to guarantee repeatability of the present method. (**A**) Variations of *b_1_* and *λ_2_* obtained by regression analysis of *β* with respect to C_dex_. (**B**) Temporal variations of *λ_b_* with respect to C_dex_. (**C**) Temporal variations of AI with respect to C_dex_.

**Figure 7 micromachines-14-00317-f007:**
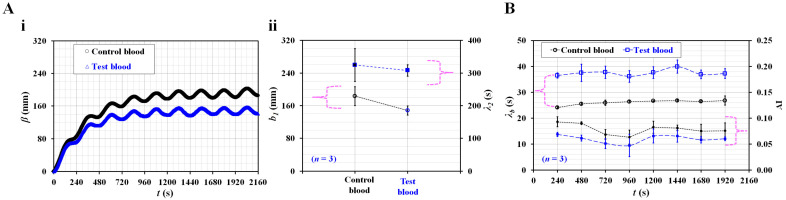
Quantitative comparison between normal RBCs and heat-shocked RBCs in terms of three rheological properties. Here, normal RBCs were exposed to a higher temperature of 50 °C for 1 h. Control blood (Hct = 50%) and test blood (Hct = 50%) were then prepared by adding normal RBCs and heat-shocked RBCs into the same concentration of dextran solution (20 mg/mL). (**A**) Contribution of heat-exposed RBCs to pressure variation. (**i**) Temporal variations of *β* with respect to control blood and test blood. (**ii**) Variations of *b_1_* and *λ_2_* with respect to test blood and control blood. (**B**) Temporal variations of *λ_b_* and AI with respect to control blood and test blood.

## Data Availability

Not applicable.

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
