# Peer review of "Biomechanical Assessment of Red Blood Cells in Pulsatile Blood Flows"

_micromachines, 2023, doi:10.3390/mi14020317_

Round 1

Reviewer 1 Report

The authors develop and test a microfluidic device capable of determining, simultaneously, the viscoelasticity, aggregation and junction pressure of blood. The author has published extensively on this topic - refs 29,32,33,35,36,45,50,51,54,53,55,62,64,67-69  (16/69 citations) - and the relation of precious work to the present study is explained in the "Introduction" to a satisfactory degree. However, it is not very clear how the results of the present study supplement those of previous work and how the derived constants compare with those found from previous methods. Given the very large body of previous work, I believe this is something that must be addressed very clearly in the "Conclusions".

One additional topic that seems to have escaped the attention of the author is the potential effect of a yield stress; some comment on this should be included in the manuscript. probably in the "Conclusions". 

The recommendation is to accept subject to satisfactorily addressing the above comments.

Author Response

Responses to the #1‘s comments

First of all, thanks for your kind and thoughtful reviews on this paper

Following your comments and suggestion, we revised the manuscript as follows;

The authors develop and test a microfluidic device capable of determining, simultaneously, the viscoelasticity, aggregation and junction pressure of blood. The author has published extensively on this topic - refs 29,32,33,35,36,45,50,51,54,53,55,62,64,67-69 (16/69 citations) - and the relation of precious work to the present study is explained in the "Introduction" to a satisfactory degree. However, it is not very clear how the results of the present study supplement those of previous work and how the derived constants compare with those found from previous methods. Given the very large body of previous work, I believe this is something that must be addressed very clearly in the "Conclusions".“ One additional topic that seems to have escaped the attention of the author is the potential effect of a yield stress; some comment on this should be included in the manuscript. probably in the "Conclusions". The recommendation is to accept subject to satisfactorily addressing the above comments.

àLike the reviewer’s comments, the author tried to suggest many methods for quantification of blood mechanical properties in a microfluidic channel. As a limitation of the previous methods, two syringe pumps were used to obtain blood viscosity, where each syringe pump was used to supply blood and reference fluid. Blood flow should be stopped for quantification of RBC aggregation. As RBC aggregation depended on hematocrit or diluent, it was necessary to monitor variations of hematocrit or diluent in terms of blood viscosity or blood pressure. As an issue, most of the previous studies were demonstrated with two syringe pumps. Blood viscosity was replaced by viscoelasticity (i.e., time constant).

àThus, the present study was aimed at reducing number of syringe pump from two to one. In addition, three biomechanical properties (i.e., RBC aggregation, time constant, and blood pressure) were simultaneously quantified under a pulsatile blood flow. As a key idea in the present study, blood flow was stopped in the side channel connected to air compliance unit. As blood was supplied at a pulsatile pattern, time constant as viscoelastic property could be obtained by analyzing blood velocity in the main channel. In addition, blood pressure at the junction point was obtained by integrating blood velocity in the side channel.

àIn principle, all formula used or derived in the study had similar expressions when compared with the previous studies. The present results showed very similar trends when compared to the previous results. However, as blood velocity obtained with micro-PIV technique had been strongly influenced by hematocrit, time constant or blood pressure calculated in terms of blood velocity had been influenced by hematocrit. Furthermore, blood pressure at the junction point was expressed as β (i.e., integration of blood velocity) because compliance constant was not available. As a future work, accurate measurement of blood velocity as well as compliance constant will be required for improving the present method.

àWith respect to yield stress, the present method did not have ability to quantify shear stress with respect to shear rate. For the reason, it was impossible to obtain yield stress existed in blood. The reviewer’s suggestion was beyond the scope of the present study. Again, the author said so sorry that enough response was not made for the reviewer’s comment.

àFollowing the reviewer’s comments, the following paragraph was newly added into the end part of “Conclusion” in the revised manuscript as,

According to the previous studies, author group was interested in quantification of blood biomechanical properties in a microfluidic channel. For quantification of blood viscosity, two high expensive and bulky size syringe pumps

were used to obtain supply blood and reference fluid, respectively. Next, for quantification of RBC aggregation, blood syringe pump was periodically turned off for stopping blood flow. Most of all, as RBC aggregation depended on hematocrit or diluent, it was necessary to monitor variations of hematocrit or diluent in terms of blood viscosity or blood pressure. In other words, most of the previous studies were demonstrated with two syringe pumps. The present study aimed to reduce number of syringe pump from two to one. Blood viscosity was replaced by blood viscoelasticity (i.e., time constant). With single syringe pump, blood flow set to pulsatile pattern. Three biomechanical properties (i.e., RBC aggregation, time constant, and blood junction pressure) were then simultaneously quantified under pulsatile blood flows. As a key idea, blood flow was stopped in the side channel connected to air compliance unit. Time constant was obtained by analyzing blood velocity in the main channel. Additionally, blood junction pressure was obtained by integrating blood velocity in the side channel. In principle, all formula used or derived in the study had similar expression when compared with the previous study. The present results showed very similar trends to the previous results. However, blood hematocrit had a strong influence on blood velocity obtained with micro-PIV technique. Furthermore, blood junction pressure was obtained as β because compliance constant was not available. Accurate measurement of blood velocity as well as compliance constant will be required for improving the performance of the present method, as a future study.

Reviewer 2 Report

Dear Editor, 

I read this manuscript and it was an interesting manuscript for me and i think it has novelty and cab be accepted. 

Best,

Author Response

Thanks for your comments. Based on the other reviewer’s comments, the original manuscript was improved.

Reviewer 3 Report

The manuscript presented the result on biophysical study of red blood cells, particularly their biomechanic properties. It is well written and data ara presented comprehensively. We recommend this manuscript for its publication 

Author Response

(The authors gave the same response as above.)
